# Street Sexual Harassment: Experiences and Attitudes among Young Spanish People

**DOI:** 10.3390/ijerph181910375

**Published:** 2021-10-01

**Authors:** Victoria A. Ferrer-Perez, Carmen Delgado-Alvarez, Andrés Sánchez-Prada, Esperanza Bosch-Fiol, Virginia Ferreiro-Basurto

**Affiliations:** 1Facultad de Psicología, Universidad de las Islas Baleares, Ctra. Valldemossa, km. 7′5, 07122 Palma de Mallorca, Spain; esperanza.bosch@uib.es (E.B.-F.); virginia.ferreiro@uib.es (V.F.-B.); 2Facultad de Psicología, Universidad Pontificia de Salamanca, C/Compañía, 1-5, 37002 Salamanca, Spain; mcdelgadoal@upsa.es (C.D.-A.); asanchezpr@upsa.es (A.S.-P.)

**Keywords:** street sexual harassment, harassment experiences, “piropos”, attitudes towards harassment

## Abstract

Violence against women (VAW) is gender-based violence directed at women and girls on account of being female that can take on multiple forms and manifest in different contexts. Among the many possible forms of VAW, this article focuses on “piropos”, a type of stranger harassment situation. Specifically, the objectives of this study were two-fold: to analyze the usefulness of a tool to evaluate social attitudes towards this form of VAW and to analyze the influence of sociodemographic variables and prior victimization (whether as a witness or victim) on attitudes towards this type of violence among Spanish youth. An opportunity sample of 538 young Spanish people took part in this study. They filled out a sociodemographic data sheet, a victimization questionnaire designed ad hoc, and a questionnaire on attitudes towards “piropos”. The results obtained indicate that the questionnaire was adequate for use as a tool to evaluate social attitudes towards this type of VAW and suggest its applicability for future studies on attitudes towards “piropos” as a type of stranger harassment situation in a Spanish context. Moreover, the results on victimization not only corroborate the magnitude of street sexual harassment in Spain and a direct effect of gender on the perception of the violence experienced, they also reinforce the need to further investigate new aspects. Regarding attitudes towards “piropos”, the results obtained indicate that, in general, participants demonstrated negative attitudes or rejection, and these feelings were particularly strong among women.

## 1. Introduction

Violence against women (VAW) is a type of gender-based violence directed at women and girls on account of being female that can take on multiple forms and manifest in different contexts [1]. Among the many possible forms of VAW, this study focuses on street sexual harassment (SSH), which constitutes a form of sexual violence [2] and, according to some authors [3,4,5,6] may also be referred to as public harassment, sexual harassment in public spaces, or stranger harassment, although others (e.g., [7]) have pointed out the existence of nuances among these different forms of harassment.

Our interest in this type of VAW is rooted in the fact that, while its precise prevalence is difficult to determine, it has been considered “*a universaling experience that almost all women share*” [8] (p. 534), and can be considered an emerging social problem in our environment given that, although it involves traditionally occurring behaviors, they have only recently been identified as problematic [3,7,9,10,11]. In fact, although feminist activism during the 1960s and 1970s focused on sexual aggression, intimate partner violence, and later on, sexual harassment (while SSH remained on the sidelines and was considered a lesser problem), this topic has become more relevant, as it forms part of the culture of oppression and continuous sexual and violent predation against women, limiting their presence in public spaces, generating fear, and constituting a precursor to physical and sexual victimization, or even femicide [3,10,12,13,14,15].

### 1.1. Conceptualisation of Street Sexual Harassment (SSH)

A significant number of studies on this subject have focused on its conceptualization [2,3,11]. Thus, in recent decades, numerous definitions have been proposed, with significant variations in the emphasis placed on the different elements that comprise SSH and their specifications [16]. As an example, Bowman [8] defined SSH as “*sexual harassment against women in public spaces made by unknown men*” (p. 51), and, more recently, SSH has been defined as “*unwanted comments, gestures, or actions forced on a stranger in a public space without their consent, directed at them because of their actual or perceived sex, gender, gender expression, or sexual orientation*” [17]. In general, all of these definitions include certain common dimensions that allow SSH to be characterized in the following way [2,3,6,8,10,16,18,19,20,21]:(a)Harassment occurs in a public or semi-public space (street, public transport) and is contextualized by a face-to-face interaction between two unknown people, that is, people who share no stable, long-term or safe connection.(b)Even though stalking, as Lopez [2] points out, is also a type of SSH, this type of violence tends to occur within a brief or even fleeting interaction (which not only constitutes one of its main characteristics, but also differentiates it from other forms of violence, such as sexual harassment in the workplace or academic environment).(c)The fleeting nature of this behavior and the source of anonymity from which it occurs hinder its evaluation and criminal prosecution [22,23].(d)The absence of an intimate or other relationship causes the behavior of the harasser to be perceived by the person harassed as an uncomfortable or even threatening transgression of her physical and psychological space.(e)The behavior is unidirectional (meaning that neither the desires nor situation of the victim are taken into account) with a singular objective (meaning it is not meant to be public nor indiscriminate).(f)The assault primarily targets women, and the instigators are primarily men (alone or in a group), although SSH may also occur against men, especially those who do not conform to a heteronormative pattern [11,24], or against the non-binary population [25].(g)Although it may be considered a benign, harmless, or even normalized and socially tolerated act, it is an act of domination that impacts the sexual freedom and right to free movement of women, communicating the message that harassers have the right to occupy public spaces and to control, assault, or injure women. It therefore assumes the imposition of the desires of one (or a few) over another (or others) and has a sexual connotation that is degrading and that objectifies, humiliates, and threatens the woman (or women), provoking in her (them) discomfort or fear.(h)It may also include visual assault (leering), nonverbal assault (sexual and obscene gestures, sighs, whistles or noises), verbal assault (jeering, sexual comments, whether supposedly positive, offensive or insulting), and/or assault in the sense of physical invasion of privacy (exhibitionism, public masturbation, groping).

### 1.2. The Incidence of the Street Sexual Harassment (SSH)

Another question that has generated considerable interest is the study of the incidence of this type of VAW. Available data corroborate Bowman’s assessment [8] of high incidence levels, as previously mentioned. As an example, Table 1 includes some results on this effect in both general terms and specific to the Spanish population, demonstrating that SSH is a serious and far-ranging social problem with a prevalence of women who have experienced it varying between 30% and 95%, with victims most frequently being younger women.

It is worth noting that incidence studies use different types of tools, with a predominance of sociological-based surveys, (i.e., [28,33,38]), although some questionnaires have been developed for this purpose (i.e., [18,34,39,40]).

### 1.3. The Types of Street Sexual Harassment (SSH)

Another aspect worth highlighting is that among the different types of SSH already mentioned, with the so-called “piropos” being among the most common [4,17,30,33,37,41]. These involve, as noted by Vallejo and Rivarola [15], and Farmer and Smock [7] or Ledezma [42], among others, verbal and nonverbal signals (including whistling, leering, comments about the person’s physical appearance, looking for attention, etc.) focused on physical aspects of the objectified women and is habitually are socially normalized and even romanticized, which could lead such behaviors to be deemed positive by society and/or by the recipient [37,41]. In general, these behaviors have no legal consequences for those who instigate them (for an analysis of this situation in Spain, see, for example, Brox [22], or Rodemann [23]), but may have behavioral or psychological effects for the recipient, including the avoidance of certain spaces or places, modifying the way they dress or their posture, or provoking negative emotions [2,4,15,20,43]. It is also worth noting that “piropos” may vary according to their intention and valence, so some behaviors may be specifically rude, vulgar, lewd, and lascivious with sexual undertones and a negative sense (more in line with the idea of catcalls), while others may have a pleasant or flattering connotation in a positive sense (more in line with the idea of flattering remarks) [37,41], and due to this distinction, a controversy arises as to whether all “piropos” are rejectable. One possible response has emerged from several studies (i.e., [37,41]), bringing to light the idea that “piropos” made by strangers generate negative effects (different, but negative in any case), regardless of whether the recipient deems them to be positive or negative. Another option could be to distinguish a “piropo” from a flattering remark with the understanding that the latter (as opposed to the former) occurs in the context of a previous interaction or relationship, is relative to a particular characteristic or quality that the recipient possesses, and is deemed to be positive by both the speaker and the listener (if the “piropo” comes from a stranger, it is intrusive and constitutes an objectification and a form of control) [2,3].

### 1.4. The Perception of Street Sexual Harassment (SSH)

In addition to analyze the incidence and most frequent typologies of SSH, other studies focused on learning how to live through an SSH experience and the psychological trauma it might produce (discomfort, fear, threat; i.e., [13,33,44]); understanding the answers and strategies that can be used to deal with these incidents (i.e., [7,15]); investigating how victims react when facing specific SSH behaviors, such as “piropo” (i.e., [37,41]); analyzing the consequences of SSH on the victim’s health (i.e., [45,46]); or determining the role of, and interventions by, witnesses to cases of VAW (i.e., [47]).

As noted by Logan [10], the majority of studies on SSH have focused on the individual level, and very few have looked at the social dimension when, in fact, this form of VAW is a social problem with social consequences that reproduce and reinforce inequalities. In this regard, it is worth recalling the fact that the use of VAW as a means to control women and limit their independence and presence in public spaces is not in the least bit new (i.e., Browmiller [48]). Indeed, SSH sends messages to the victims as well as the aggressor, and even to the witnesses “*about the power, the violence, the equality, the civic responsibility, the public space, and the freedom*” [10] (p. 206). This suggests that SSH is not only a form of VAW and sexual violence but also a symbolic violence in that it brings to light the limitations that a patriarchal society imposes on the freedom and mobility of women and restricts their use of public spaces at certain times and places [2,3,4,19,43].

### 1.5. The Beliefs and Attitudes towards Street Sexual Harassment (SSH)

To this end, it is important to remember that the beliefs and attitudes towards VAW (in general and in its different manifestations) play an important role in the perpetration of these types of violence and in the response of the women who suffer from these experiences as well as the communities in which they occur [49,50]. According to a previously described proposition by Logan [10], the study of social attitudes towards SSH is particularly relevant, because it determines the social norms towards this type of violence and the formal and informal responses it evokes, such that a climate of acceptance and favorable and/or tolerant beliefs and attitudes becomes a risk factor for the proliferation of SSH.

These attitudes may be influenced by different factors, including the victim’s, or even the harasser’s, sociodemographic characteristics or the effects of the context or of sexism [18,38,51,52]. This is why, for example, the fact that SSH is primarily endured by women could have a direct (greater perception of this violence by women) (i.e., [53]) or indirect (mediated by the gender role and/or sexism) (i.e., [18,54] effect on the perception of the violence endured by the victim.

### 1.6. Study Objectives

Given the relevance of SSH and “piropos” in general, and particularly in the Spanish context [41], and given the still-limited volume of empirical studies on the matter in Spain (see Table 1), this study focused on “piropos” as a type of stranger harassment situation. Its objectives were two-fold: to analyze the usefulness of a tool to evaluate social attitudes towards this form of VAW and to analyze the influences of sociodemographic variables and prior victimization (whether as a witness or victim) on attitudes towards this type of violence among Spanish youth.

## 2. Materials and Methods

### 2.1. Participants

An opportunity sample of 538 Spanish young people aged between 18 and 30 years (M = 21.51, D.T. = 2.70) took part in this study. Regarding gender, all participants in the study sample categorized themselves as either female or male, of which 397 (73.8%) were female and 141 (26.2%) were male. Their characteristics are presented in Table 2.

Based on the age variable, two groups were created: individuals aged 18 to 21 years (late adolescents or youths; *n* = 307; 57.1%) and those aged 22 to 30 years (young adults; *n* = 231; 42.9%). This division took into account life expectancy and the age range of the participants and essentially corresponds to the difference between Generation Z or Centennials (born between 1995 and 2009) [55] and Generation Y or Millennials (born between 1981 and 1994) [56]. The distribution between males and females in these groups was homogeneous: 75.6% of girls in the 18–21 age group and 71.4% in the 22–30 age group (χ^2^ (1, *n* = 538) = 0.322, *p* = 0.163).

### 2.2. Instruments

Data collection was accomplished via a questionnaire that included the following: A sociodemographic data sheet with information related to gender (self-categorized by participants), age, economic level, current work status, and political opinion.A victimization questionnaire designed ad hoc, including eight questions with a 4-point scale answer relative to the frequency (1: No, never; 2: Yes, on one occasion; 3: Yes, on more than one occasion; and 4: Yes, regularly) with which the participants had been victims of (4 questions) or witness to (4 questions) 4 types of violence: robbery, SSH, sexual harassment, and intimate partner violence.A questionnaire on attitudes towards “piropos” [40] to measure attitudes towards a SSH situation: “piropos”. The following situation was presented: a young girl is walking alone down a street and receives the sexual attention of a group of young males, specifically a comment from one of them about ‘how hot she is’. The participants were asked to evaluate their perception of this situation, indicating their level of agreement according to a Likert scale (where 1 means ‘Not at all’ and 7 means ‘A lot’) regarding 7 items characterizing the situation (as fun, pleasant, flattering, male chauvinist, offensive, unpleasant, vulgar). For items 1, 2, and 3, the order of the scoring scale was inverted prior to the analysis so that higher scores reflected a negative attitude or rejection of the remark. According to the author, the scale offers good internal consistency (α = 0.91).

### 2.3. Procedure

A non-probability convenience sample was used. The questionnaire used to gather information was prepared on the Lime Survey platform and disseminated through social network sites used by the researchers and their collaborators. An introductory text about the objectives and conditions of the study was included, and access to the answer sheet implied prior agreement of the participants to take part in the study.

### 2.4. Data Analysis

The psychometric adjustment of the Questionnaire on Attitudes towards “piropos” in this sample was verified with a reliability analysis using Cronbach’s alpha and an Exploratory Factor Analysis based on the principal axis factoring method to test its unidimensionality, as proposed in Moya-Garofano’s original version [40]. This was followed by a descriptive analysis of the frequency with which the types of victimization under study appeared, and a chi-squared test was applied to explore the relationship with gender. ALSCAL (Alternating Least Square Analysis) Multidimesional Scaling was applied to explore the relationships among the different types of victimization and to identify a possible underlying substructure. In order to study differences regarding the frequency of the different types of victimization studied and the attitudes towards stranger harassment according to gender and age, the Student’s *t*-test for independent samples was used, after previously confirming the homoscedasticity of each variable with the Levene test for homogeneity of variance. If the measure of homoscedasticity (*p* > 0.05) was met, the *t*-value for the homogeneity of variance was used, and when it was not met, the value of t was taken with non-homogenous variance once the degrees of freedom had been corrected. In order to identify which variables of victimization (as a witness or victim) were predictive of attitudes towards stranger harassment among men and women according to age, a stepwise multiple regression analysis was performed. Finally, the differences in attitudes towards stranger harassment based on the other sociodemographic variables under study were analyzed using ANOVA. If the measure of homoscedasticity (*p* > 0.05) (determined via the Levene test) was not met, a contrast post-hoc test was applied to the different groups using the Games–Howel test for non-homogeneous groups. It should be noted that, due to the small size of some subsamples, some variables were recoded. Socioeconomic status was recoded into three categories (low and lower middle, middle-middle, and upper middle and upper). Regarding political ideology, only three categories were compared: left, right, and center. Regarding level of education, participants with primary level education were not taken into account.

## 3. Results

### 3.1. Psychometric Properties of the Questionnaire on Attitudes towards “Piropos”

The internal consistency of the scale as a whole was satisfactory with a Cronbach’s alpha value of 0.884. The Exploratory Factor Analysis based on the principal axis factoring method (after inverting the scores for items 1 to 3) yielded a solution with a non-rotation matrix, accounting for 69.48% of variance and a general factor (Factor 1) accounting for 56.43% of variance. The factorial load is presented in Table 3. This result confirms the unidimensionality of the scale, which was deemed to be adequate to obtain a general score from the sum of the items (inverting the scores for items 1 to 3), as formulated in the original version. Using this scale, the total possible score of attitudes towards “piropos” varied between 7 and 49, with higher scores reflecting more negative attitudes or rejection of this type of stranger harassment situation.

### 3.2. Frequencies of the Different Types of Victimization and Relationship with Gender

The frequency distribution, as shown in Table 4, indicated a clear pattern based on gender. An analysis of the relationship between type of victimization and gender (based on a chi-squared test) showed that, while the frequency with which the participants were witnesses and/or victims of theft was independent of their gender, there was a statistical relationship between these two variables for the other forms of victimization analyzed, indicating that, in all cases, women had more frequently experienced being a witness or victim.

### 3.3. Relationships among Different Types of Victimization

Given the previously described results and the unequal sizes of the subsamples of women and men, ALSCAL Multidimensional Scaling was applied to each subsample separately to explore the relationships between the frequencies of different types of victimization, with witnesses or victims assessed separately in both subsamples. The results obtained indicate a clear underlying substructure among the women but not among the men.

Specifically, a bidimensional structure with excellent fit values was obtained from the sample of females (S-stress = 0.026, RSQ = 0.997). Figure 1 shows that theft (as either victim or witness) constitutes an obviously different form of victimization, while the grouping of intimate partner violence and sexual harassment emerged from direct experiences as a victim. SSH (whether as witness or victim) constitutes a different grouping from other types of victimization.

On the other hand, a different structure with poor fit values was obtained from the sample of males (Figure 2) (S-stress = 0.085, RSQ = 0.970), which does not allow for clear interpretation of the configuration.

### 3.4. Victimization and Attitudes towards “Piropos” by Gender

An analysis of the differences in victimization by gender (Table 5, Figure 3) showed that there were no statistically significant differences between men and women with regard to being a victim of theft; men were witnesses of theft with a higher statistically significant frequency than women, and for the three forms of VAW studied (intimate partner violence, sexual harassment and SSH), women had been a witness or victim with a significantly higher frequency than men. Regarding attitudes towards “piropos”, both men and women presented higher than average scores (greater than 42.70 points from a maximum of 49 points), which implies negative attitudes towards, or rejection of, such actions, although women showed significantly greater rejection of this type of SSH situation than men (Table 5).

### 3.5. Victimization and Attitudes towards “Piropos” by Age

An analysis of the differences in victimization by age (see Table 6, Figure 4) showed that the only statistically significant differences were related to victimization by theft, as either a witness or victim, with both occurrences significantly more frequently among participants aged 22 to 30 than among those aged 18 to 21.

Regarding attitudes towards “piropos”, both age groups presented higher than average scores (greater than 44.50 points from a maximum of 49 points), implying negative attitudes or rejection of this type of SSH situation with no observable statistically significant differences between both groups (Table 6).

### 3.6. Relationship between Victimization and Attitude towards “Piropos” as a Function of Gender and Age

According to the analyses performed by gender, among men, no variables of victimization entered the regression equation, which led us to conclude that none of the victimization variables under study were statistically significantly predictive of their attitudes towards “piropos”. Among the women, the act of witnessing SSH was the only variable of victimization that was shown to serve as a statistically significant predictor of attitudes towards “piropos” with a direct relationship (change in R^2^ = 0.025; *p* = 0.004; β = 0.158); in other words, having been witness to SSH predicted greater rejection of a SSH situation, such as “piropos”.

By age, the victimization variables that were found to be statistically significant predictors of attitudes towards “piropos” among the younger participants (aged 18 to 21) were witnessing SSH (change in R^2^ = 0.036; *p* = 0.004; β = 0.142), theft (change in R^2^ = 0.031; *p* = 0.007; β = −0.227), or intimate partner violence (change in R^2^ = 0.031; *p* = 0.006; β = 0.195). In other words, witnessing SSH or intimate partner violence is predictive of greater rejection of “piropos” (direct relation), while witnessing theft is predictive of lesser rejection of “piropos” (inverse relation). Among participants aged 22 to 30, the only predictive victimization variable of attitudes towards “piropos” was having been a victim of SSH (change in R^2^ = 0.022; *p* = 0.025; β = 0.150); that is, having experienced SSH is predictive of greater rejection of this type of SSH situation.

### 3.7. Attitudes towards “Piropos” and Other Sociodemographic Variables

In general terms, all subgroups analyzed as functions of the different categories under study had above average point scores (greater than 40.50 points of a maximum 49 points) on the scale of attitudes towards “piropos”, which suggested the presence of negative attitudes or rejection (Table 7). Regarding observable differences, although the F contrast statistic was significant for the work status and socioeconomic status, there were no significant differences in any of the respective groups for each variable when comparing them and correcting for the effect of homogeneity of variance. The only variable that was suggested to have significant effects on the groups was political ideology. Specifically, those who claimed a more ‘left-leaning’ affiliation scored more negatively or had a greater rejection of “piropos” than those who claimed a more ‘center’ or ‘right-leaning’ affiliation.

## 4. Discussion

The results obtained indicate the adequacy of the Questionnaire of Attitudes towards “piropos” [40] as a tool to specifically evaluate social attitudes towards this type of VAW in the study sample within a Spanish context, suggesting their adequate reliability and supporting the author’s use of a unidimensional point scale. It should be noted that an important type of tool available for analyzing SSH is sociological-based surveys [28,33,38]; therefore, the availability of a questionnaire suitable for this purpose could be very useful for research on the topic. Moreover, although some related studies [18,34,39] report evaluate attitudes and beliefs regarding SSH using questionnaires, a review of their content and contributions showed different objectives (such as to determine which components of SSH are perceived by women who have experienced them, or to estimate the frequency of SSH), and demonstrated they were developed in different cultural contexts outside of Spain (such as Venezuela, Paraguay or Peru). In this sense, the results regarding the adequacy of the Questionnaire of Attitudes towards “piropos” are not only promising in this study but also suggest its applicability in future studies on attitudes towards this type of SSH situation performed in Spain.

Regarding the experience of victimization, it should be noted, first of all, that nearly 8 out of every 10 participants in the sample study (79%) had been victims of SSH on at least one occasion, with a notable difference between men (somewhat less than half, 42.6%) and women (nearly all participants, 94.7%). If we look not only at the total number but at the frequency, a clear pattern by gender can be observed, since SSH was a common experience for nearly one-third of the women (29.5%) studied while affecting less than 1% of men (0.7%). The relation between direct experience of SSH and gender detected in the sample can be extended to the general public with a probability error of less than 0.1% (*p* < 0.001). These data not only provide an idea of the magnitude of the problem and corroborate the results obtained in previous studies carried out in Spain (i.e., [38]), they also show similar results to those previously obtained in countries such as Canada [26], the USA [9], Nicaragua [29], Chile [33], and Peru [34].

In addition, more than 8 out of every 10 participants (84.8%) had witnessed this form of VAW on at least one occasion. This case study also identified a significant relationship with gender, given that 69.9% of males had witnessed VAW on at least one occasion as opposed to 89.9% of women (35% on a regular basis). This result not only corroborates the magnitude of the problem, it reinforces the need to further investigate new aspects, such as the behavior of those who witness this form of violence, in order to, at that develop the necessary tools to induce a proactive response to avoid and/or minimize such violence [47].

Beyond the proportion of people who had witnessed or directly experienced SSH, the ALSCAL Multidimensional Scaling used in this study provided interesting suggestions regarding the experiences of victimization endured by female participants. Specifically, the results clearly indicate that for them, theft constitutes a specific and different type of victimization, while the other types of violence under study (intimate partner violence, sexual harassment and SSH) are different, thus marking the distinctiveness of these cases as forms of VAW [1,2,3,4,6]. Additionally, being a victim of VAW by a partner or being a victim of sexual harassment is relevant to define a clear type of victimization, while in the case of SSH, the resulting group included those who had been victims or witnesses to the occurrence of SSH. It is important to remember that Multidimensional Scaling offered a very clear configuration for the subsample of women (and not for men). It stands to hypothesize that this result may be related to the characteristics of this subsample, which consisted of young women (less than 30 years of age), corresponding to the typical profile (young woman) of the primary group of victims (directly, but also as witnesses to assaults on others in their peer group) of this form of VAW [15,17,28,32,34,35]. In fact, the results obtained regarding victimization by gender and age were in accordance with this, indicating that women are witnesses to, and victims of, the different forms of VAW analyzed in this study significantly more often than men, with no difference between the age groups studied.

Regarding the effects of gender on the perception of the violence experienced, we emphasize that the results described in the literature on this topic are contradictory. While some studies have indicated the existence of a direct effect, whereby women would identify more types of behavior as forms of VAW (i.e., [53,57]), other studies indicate that gender itself does not have a direct effect, although an indirect effect, mediated by factors such as the role of gender and/or sexism, seem to exist [18,54]. The results found by Vallejo and Rivarola [15] provide a complementary explanation, noting a gradient in the perception, tolerance, and justification of SSH whereby, although the majority of women consider behaviors such as leering, catcalls, kissing sounds, or other comments to cause general unease, these behaviors are only identified as actual violence when they involve specific a sexual reference. However, in the specific case of individuals who had been a victim or witness to SSH, the results obtained in this study from the previously discussed Multidimensional Scaling suggest the existence of a more direct relationship between having a first-hand experience (personally or as a witness) of an incident involving SSH and the ability to identify it as such. Thus, further in-depth analysis of this matter is required in order to draw more definitive conclusions and determine possible applications to other forms of VAW. Along these lines, one relevant aspect for future investigation was noted by Delgado [12] regarding the possible effect of the presence/absence of a feminist discourse among the participants about their distinction when identifying and explaining the dynamics of assault and violence endured both in general and within an urban context.

Regarding attitudes towards “piropos”, the results obtained indicate that, in general, the participants in this study demonstrated negative attitudes or rejection, which was particularly strong among women, towards these types of remarks. In fact, as expected, and as in the case of other forms of VAW [49,50,58,59,60], gender accounted for statistically significant differences, with women displaying much more negative attitudes towards “piropos”, interpreting them as a form of VAW. A general rejection of this form of VAW was observed by age, similar to what has been described in the literature for other forms of this violence among persons of a similar age [60], but no significant differences were observed regarding attitudes towards “piropos” between the two age groups compared.

In terms of the possible role of victimization in the prediction of attitudes towards “piropos”, witnessing SSH was predictive of greater rejection of “piropos” among women and younger participants and having been a victim of SSH was predictive of greater rejection towards “piropos” among persons between the ages of 22 and 30. This result seems to fall in line with the result sof studies carried out by Moya-Garofano et al. [37,41] in the sense that having directly received or witnessed SSH constitutes a negative life experience that an individual has endured, causing them to reject it. Further in-depth analysis of the results is required to determine the specific effects of different forms of victimization and other SSH situations.

Finally, a comparison based on other sociodemographic variables demonstrated that participants with a more left-leaning political ideology displayed a significantly greater level of rejection towards “piropos” (while those with a more right-leaning political ideology displayed the highest levels of acceptance of the subgroups analyzed). It is worth noting that previous studies highlighted effects of political affiliation on attitudes towards sexual conduct [61] and towards other forms of VAW [58], similar to the results obtained in this case and related to a more traditional understanding of the social roles of women and men and the gender mandates among individuals with more right-leaning political tendencies.

It is worth noting that this study is not without limitations. Among others is the fact that it involved a convenience sample that was primarily composed of women and young people (below the age of 30). As such, it is necessary to continue working to determine the point at which these results can be applied to other populations, especially among different age cohorts. Additionally, the use of a single questionnaire that only evaluated the attitudes towards one type of SSH situation (“piropos”) is also considered a limitation of this study, although it could be compensated in the future by establishing comparisons between the results obtained with the applied questionnaire and others designed for the evaluation of perceived SSH (i.e., [39]). Likewise, further research on different aspects of SSH is required, such as, research on its components and, in particular, the possible differences in attitudes towards vulgar “piropos” (catcalls) or (supposedly) positive “piropos” [41].

Despite these limitations, the results obtained represent a notable step in achieving greater understanding of this topic that is socially relevant in our current environment. It provides a better understanding of a possible evaluation tool as well as a better understanding of the attitudes among different subsamples and their relationships with different victimization experiences.

## 5. Conclusions

This study allowed us to study “piropos”, a type of stranger harassment situation, to fulfill two objectives: to determine the usefulness of a tool to evaluate social attitudes towards this form of VAW and to identify the influences of sociodemographic variables and prior victimization on attitudes towards this type of violence among the Spanish youth population.

Our results demonstrate the adequacy of the questionnaire used and its applicability for future studies on this topic. Regarding the experience of victimization, the results obtained by gender and age indicate that women are witness to, and victims of, the different forms of VAW analyzed in this study significantly more often than men, with no difference between the age groups studied. These results also suggest that there is a direct effect of gender on the perception of the violence experienced with a direct relationship between first-hand experience (personally or as a witness) of an incident involving SSH and the ability to identify it as such. In general, the participants in this study demonstrated negative attitudes or rejection towards “piropos” with no significant differences between the two age groups, but attitudes were particularly strong among women and participants with a more left-leaning political ideology.

These results have important practical and methodological implications. From a practical point of view, the results obtained represent a notable step towards achieving a greater understanding of this topic that is socially relevant in our current environment. Our results provide a better understanding of the attitudes among different subsamples and their relationships with different victimization experiences. Particularly relevant from the point of view of VAW prevention is the high proportion of people, especially women, who had witnessed SSH. This means, on the one hand, that it is important to train men to identify this form of violence when it occurs in public sites, and on the other hand, means that it is important to carry out programs to encourage the active intervention of witnesses to this violence to prevent it. From a methodological point of view, the results obtained provide new evidence on the use of tools to assess attitudes towards different forms of VAW.

## Figures and Tables

**Figure 1 ijerph-18-10375-f001:**
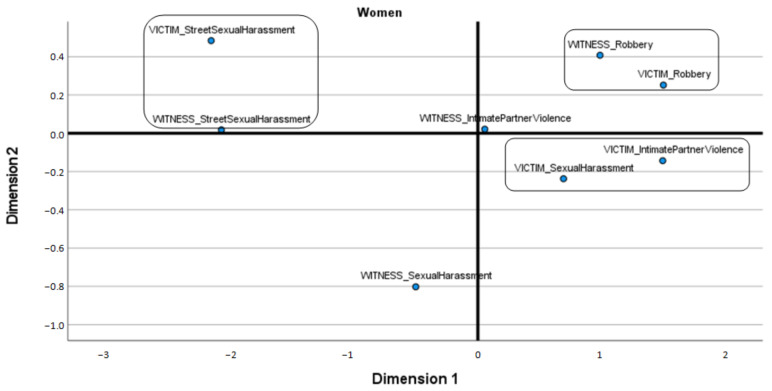
Multidimensional scaling of victimization types in women.

**Figure 2 ijerph-18-10375-f002:**
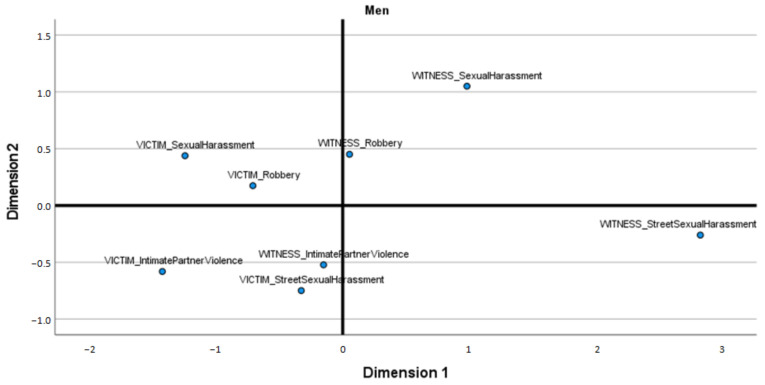
Multidimensional scaling of victimization types in men.

**Figure 3 ijerph-18-10375-f003:**
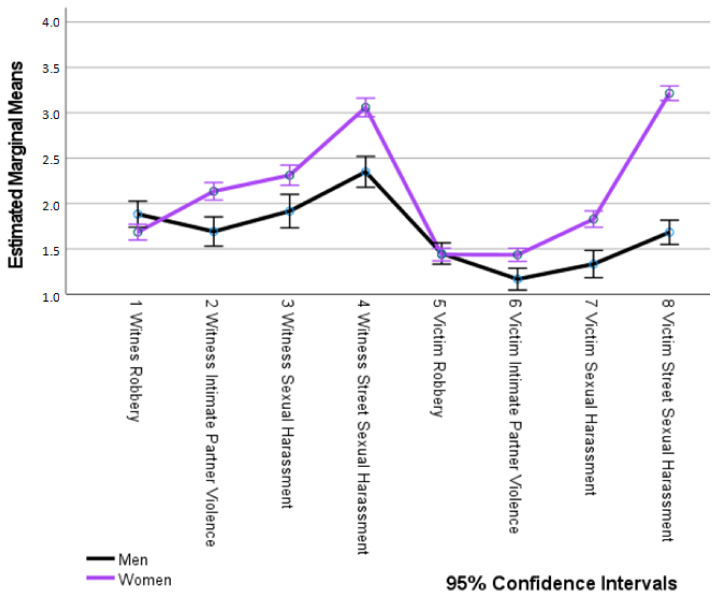
Differences in victimization types by gender.

**Figure 4 ijerph-18-10375-f004:**
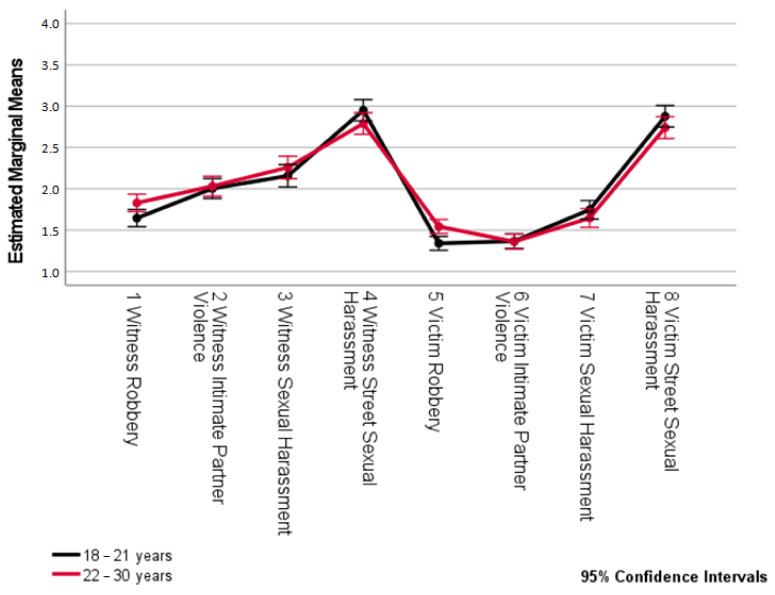
Differences in victimization types by age.

**Table 1 ijerph-18-10375-t001:** SSH (street sexual harassment) incidence.

Country and Year of the Study	Source	Method	Results
Canada, 1993	MacMillan et al. [26]	General female population survey	More than 80% of women surveyed had experienced SSH.
USA, 2014	Kearl [9] Stop Street Harassment [27]	Sociological survey	SSH is a significant problem in the country. It has been experienced by 65% of women, especially in the forms of verbal (57%) and physical (41%) harassment.
Foundation Jean Jaurés multi-country study, 2018—USA	Clavaud et al. [28]	Survey on sexual violence and street harassment	Fifty-seven percent of women (especially young women) have experienced at least one SSH situation in their lifetime: 70% whistling; 50% sexist comments, teasing or insults; 46% rude gestures with sexual connotations; 41% touching in a sexual way without their consent.
Nicaragua, 2014	Gutierrez & Lovo [29]	Sociological survey	More than 95% of women interviewed had experienced at least once situation of SSH, especially through gestures or verbal language and on the street, in markets, or on public transport. Nearly 41% had experienced a strong SSH experience, (experiences that, because of their severity and context, were remembered as causing more fear, anger, or frustration, and/or as leading to changes in their routines or behaviors).
Mexico, Mexico City, 2013	Campos et al. [30]	Survey	A total of 62.8% of women who sought services at Mexico City Government Health Secretariat clinics reported having experienced some type of SSH in the previous month.
México, Queretaro, 2014	Meza & García [31]	Survey of adolescents aged 13–15	Almost half of the adolescents had experienced SSH situations, almost 70% of girls had experienced SSH.
Uruguay, 2013	UN-Women [32]	Sociological survey about VAW	More than 50% of women aged 15–29 had experienced some form of sexual harassment in public spaces in the previous year.
Ecuador, Quito, 2012	UN-Women [32]	Survey on the Quito Safe City Programme	More than 65% of women had experienced some form of sexual harassment in the city, often on public transport.
Chile, 2014	OCAC [33]	Sociological survey	A total of 99.4% of women experienced SSH at some point, almost 40% daily, and 77% at least once a week.
Perú, 2012	Vallejo & Rivarola [15]	National Survey on Family and Gender Roles. Public Opinion Institute of the PUCP	SSH practices mainly affected young women and female students.
Perú, Lima, 2016	Llerena [34]	Questionnaire administered to female university students	Ninety-one percent had experienced at least one SSH situation in the previous year.
Multi-country study, 2018	Plan Internacional [17]	Social survey tool on maps in Madrid, Delhi, Kampala, Lima, and Sidney	There was a significant volume of SSH complaints perpetrated by groups on girls and young women in the five cities analyzed, especially on the street and on public transport.
Fondation Jean Jaurés multi-country study, 2018—Europe	Clavaud et al. [28]	Survey on sexual violence and street harassment in France, Germany, Spain, Italy, and United Kingdom	Fifty-six percent of women surveyed (especially young women) had experienced at least one SSH situation in their lifetime: 65% whistling; 36% sexist comments, teasing or insults
United Kingdom, London, 2012	UN-Women [35]	No information	Forty-three percent of young women had experienced SSH in the previous year.
France, 2013	UN-Women [35]	No information	Twenty percent of women had experienced SSH in the previous year.
Spain, 2016	Rodríguez, et al. [5] Rodríguez, et al. [36]	Analysis of tweets on the topic	Around 75% of the harassment situations described had occurred in public places, mainly to young women and in the street.
Spain, 2016	cited in Moya-Garofano et al. [37]	Questionnaire administered to female university students	A total of 25.9% of participants had experienced SSH through rude catcalling 2–4 times a month, 17.8% at least once a month, and only 5.5% had never experienced catcalling.
Fondation Jean Jaurés multi-country study, 2018—Spain	Clavaud et al. [28]	Survey on sexual violence and street harassment	Fifty-five percent of women (especially young women) had experienced at least one SSH situation in their lifetime: 86% whistling; 76% insistent looks; 50% rude gestures with sexual connotations; 44% insistent approaches without consent, and 40% following, or sexist comments, teasing or insults.
Spain, 2018	Varela, Caja & Rueda [38]	Survey of women aged 14 to 66 years old	Ninety-nine percent had experienced SSH: 32% occasionally, 31% monthly, 25% weekly, 12% daily.

VAW: Violence Against Women; PUCP: Pontificia Universidad Católica de Perú.

**Table 2 ijerph-18-10375-t002:** Sample description.

Variable	Classification	*n* (%)
Educational level	Primary	3 (0.6%)
	Secondary	310 (57.6%)
	Professional training	64 (11.9%)
	University	161 (29.9%)
Labor situation	Unemployed	38 (16.5%)
	Employed	139 (25.8%)
	Students	360 (66.9%)
	No reply	1 (0.2%)
Economical level	Low	89 (16.5%)
	Medium-low	155 (28.8%)
	Medium-medium	242 (45.0%)
	Medium-high	50 (9.3%)
	High	2 (0.4%)
Political opinion	Left	371 (69.0%)
	Centre	106 (19.7%)
	Right	35 (6.5%)
	Others	1 (0.2%)
	No reply	23 (4.3%)
Current partner	Yes	79 (14.7%)
	No	452 (84.0%)
	No reply	7 (1.3%)

**Table 3 ijerph-18-10375-t003:** Factor loadings of the Questionnaire on Attitudes towards “Piropos”.

Items	Unrotated Matrix	
	Factor 1	Factor 2
1. It is fun *	0.696	0.499
2. It is nice *	0.599	0.515
3. It is flattering *	0.699	0.318
4. It is sexist (macho)	0.738	−0.211
5. It is offensive	0.800	−0.298
6. It is unpleasant	0.846	−0.313
7. It is disgusting	0.848	−0.259

* Recoded scores.

**Table 4 ijerph-18-10375-t004:** Frequency of victimization by gender.

Type	Frequency	Gender	Witness	Victim	*χ*^2^ (3 df)
Robbery *n* = 453	Never	Men	50 (41.7%)	75 (62.5%)	Witness χ^2^ = 7.37 *p* = 0.061 Victim χ^2^ = 1.56 *p* = 0.459
		Women	171 (51.4%)	219 (65.8%)	
	Once	Men	35 (29.2%)	36 (30.0%)	
		Women	96 (28.8%)	82 (24.6%)	
	More than once	Men	34 (28.3%)	9 (7.5%)	
		Women	66 (19.8%)	32 (9.6%)	
	Usually	Men	1 (0.8%)	-	
		Women	-	-	
Intimate Partner Violence *n* = 537	Never	Men	83 (58.9%)	124 (87.9%)	Witness χ^2^ = 38.75 *p* < 0.001 Victim χ^2^ = 17.30 *p* = 0.001
		Women	122 (30.8%)	279 (70.3%)	
	Once	Men	20 (14.2%)	11 (7.8%)	
		Women	112 (28.2%)	67 (16.9%)	
	More than once	Men	30 (27.0%)	6 (4.3%)	
		Women	145 (36.6%)	48 (12.1%)	
	Usually	Men	-	-	
		Women	17 (4.3%)	2 (0.5%)	
Sexual Harassment *n* = 537	Never	Men	64 (45.4%)	110 (78.0%)	Witness χ^2^ = 22.62 *p* < 0.001 Victim χ^2^ = 38.10 *p* < 0.001
		Women	124 (31.2%)	191 (48.2%)	
	Once	Men	24 (17.0%)	15 (10.6%)	
		Women	37 (9.3%)	92 (23.2%)	
	More than once	Men	49 (34.8%)	16 (11.3%)	
		Women	196 (49.5%)	106 (26.8%)	
	Usually	Men	4 (2.8%)	-	
		Women	39 (9.8%)	7 (1.8%)	
Street Sexual Harassment *n* = 537	Never	Men	42 (29.8%)	81 (57.4%)	Witness χ^2^ = 57.39 *p* < 0.001 Victim χ^2^ = 230.5 *p* < 0.001
		Women	40 (10.1%)	21 (5.3%)	
	Once	Men	18 (12.8%)	25 (17.7%)	
		Women	30 (7.6%)	23 (5.8%)	
	More than once	Men	70 (49.6%)	34 (24.1%)	
		Women	187 (47.1%)	235 (59.3%)	
	Usually	Men	11 (7.8%)	1 (0.7%)	
		Women	139 (35.0%)	117 (29.5%)	

**Table 5 ijerph-18-10375-t005:** Victimization and attitudes toward “piropos” by gender.

Variable	Lévène (*p*)	Gender	*n*	Mean (*SD*)	*df*	*t* Student	*p*
Witness to Robbery	0.399	Men	120	1.88 (0.852)	471	2.326	0.020
		Women	333	1.68 (0.784)			
Witness to Intimate Partner Violence	0.716	Men	141	1.68 (0.873)	535	−5.246	<0.001
		Women	396	2.14 (0.910)			
Witness to Sexual Harassment	0.088	Men	141	1.95 (0.959)	535	−4.316	<0.001
		Women	396	2.38 (1.030)			
Witness to Street Sexual Harassment	0.000	Men	141	2.35 (0.994)	228.779	−7.536	<0.001
		Women	396	3.07 (0.910)			
Victim of Robbery	0.673	Men	120	1.45 (0.633)	451	0.169	0.868
		Women	333	1.44 (0.663)			
Victim of Intimate Partner Violence	0.000	Men	141	1.16 (0.472)	376.230	−4.902	<0.001
		Women	396	1.43 (0.720)			
Victim of Sexual Harassment	0.000	Men	141	1.33 (0.673)	323.972	−6.756	<0.001
		Women	396	1.82 (0.889)			
Victim of Street Sexual Harassment	0.000	Men	141	1.68 (0.865)	217.650	−17.734	<0.001
		Women	396	3.13 (0.741)			
Attitudes toward “piropos”	0.000	Men	139	42.72 (7.73)	207.499	−4.243	<0.001
		Women	396	45.82 (6.73)			

**Table 6 ijerph-18-10375-t006:** Victimization and attitudes toward “piropos” by age.

Variable	Lévène (*p*)	Age	*n*	Mean (*SD*)	*df*	*t* Student	*p*
Witness to Robbery	0.581	18–21	229	1.65 (0.785)	451	−2.443	0.015
		22–30	224	1.83 (0.819)			
Witness to Intimate Partner Violence	0.478	18–21	306	2.00 (0.934)	535	−0.646	0.519
		22–30	231	2.05 (0.907)			
Witness to Sexual Harassment	0.874	18–21	306	2.25 (1.031)	535	−0.295	0.768
		22–30	231	2.28 (0.1027)			
Witness to Street Sexual Harassment	0.355	18–21	306	2.95 (0.976)	535	1.805	0.072
		22–30	231	2.80 (0.990)			
Victim of Robbery	0.000	18–21	229	1.34 (0.590)	434.971	−3.348	0.001
		22–30	224	1.54 (0.701)			
Victim of Intimate Partner Violence	0.682	18–21	306	1.35 (0.677)	535	−0.311	0.756
		22–30	231	1.37 (0.672)			
Victim of Sexual Harassment	0.693	18–21	306	1.71 (0.860)	535	0.405	0.685
		22–30	231	1.68 (0.871)			
Victim of Street Sexual Harassment	0.008	18–21	306	2.77 (1.063)	524.602	0.465	0.636
		22–30	231	2.73 (0.923)			
Attitudes toward “piropos”	0.163	18–21	305	45.35 (6.149)	533	1.329	0.185
		22–30	230	44.56 (7.730)			

**Table 7 ijerph-18-10375-t007:** Attitudes toward “piropos” and other socio-demographic variables.

Variable	Classification	*n*	Mean	*SD*	*F*	*p*	G.-H *
Educational level	Secondary	307	45.03	6.97	F(2.529) = 0.850	0.428	----
	Professional training	64	44.08	8.01			
	University	161	45.40	6.20			
Labor situation	Unemployed	38	42.47	9.88	F(2.529) = 4.325	0.014	N.S. **
	Employed	138	44.33	7.73			
	Students	358	45.53	6.04			
Economical level	Low/Medium-low	243	44.94	7.20	F(2.532) = 3.402	0.034	N.S. **
	Medium-medium	241	45.55	5.84			
	Medium-high/High	51	42.80	9.20			
Political opinion	Left	369	46.49	5.48	F(2.506) = 24.774	<0.001	Left-Centre Left-Right ***
	Centre	105	42.82	8.19			
	Right	35	40.54	7.69			
Current partner	Yes	79	45.44	5.701	F(1.529) = 0.346	0.557	
	No	452	44.95	7.076			

* *p*—significance of Games–Howel post-hoc test for non-homogeneous groups. ** N.S.: non-significant difference between a pair of groups. *** Significant differences between Left—Centre (*p* < 0.001) and Left—Right (*p* < 0.001). No differences between Centre—Right opinion (*p* = 0.302).

## Data Availability

The raw data supporting the conclusions of this manuscript will be made available by the authors on request.

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
