# Peer review of "Street Sexual Harassment: Experiences and Attitudes among Young Spanish People"

_ijerph, 2021, doi:10.3390/ijerph181910375_

Round 1

Reviewer 1 Report

This article will make an excellent contribution to the literature on the topic of street sexual harassment (piropos).  However, I have a suggestion to make the paper even stronger: the authors could incorporate a brief addition that explores the psychosocial basis for piropos. This in turn would help readers begin to address adverse consequences of the custom, since the journal's concern with public health requires more coverage of what we do with the data collected.  That is, how do we address this inculcated cultural practice?  In order to provide some insight on that (to at least begin to address how to act on the study's findings), the authors need greater consideration of "socio-cultural" factors (delineated in IJERPH's Aims and Scope). 

I suggest that the authors incorporate the following article co-authored by Marcelo M. Suárez-Orozco:

Suárez-Orozco, Marcelo M. and Alan Dundes. "The piropo and the dual image of women in the Spanish-speaking world." Journal of Latin American Lore 10, no. 1 (1984): 111-133.

Note: This same article was re-printed in Alan Dundes, Parsing Through Customs (Madison: University of Wisconsin), 1987, on pages 118-144.

Note: On his Wikipedia page, it states that:

Marcelo Suárez-Orozco is the ninth permanent and current chancellor of the University of Massachusetts Boston and the first Latino to lead a campus in the Massachusetts public university system.

I hope that the authors can find some insight from this article that will help them go beyond documenting the problem so that greater coverage about the socio-cultural implications of the phenomenon can help promote positive social change.

Overall, the article is very well done.

I found a couple of typos:

L 33: Cut the word “a”: Violence against women (VAW) is a gender-based violence

L: 104: There is stray punctuation:     ).

Author Response

Dear Reviewer,

According to your suggestions, we have revised and changed our paper “Street sexual harassment: experiences and attitudes among young Spanish people” (ijerph-1399638) in order to improve it.

Specifically we have made the following changes:

  • We have corrected the typographic mistakes pointed out.
  • We have reviewed Suarez-Orozco and Dundes paper recommend. And, related to the suggestions about including some other points of view, although the suggestions of deepen on the socio-cultural basis of the “piropos” and incorporate the anthropological perspective included in the Suarez-Orozco and Dundes paper is really very interesting, we have considered that it exceeded the objectives of the present paper, centred specifically in the analysis and assessment of a psychosocial topic such as the attitudes toward this type of violence against women. In any case, we very appreciate these suggestions and we will include this topic and this reference in our future papers about "piropos".

We remain at your disposal for any further modifications or adjustments that you consider relevant.

Yours sincerely

The authors

Reviewer 2 Report

This manuscript addresses a very relevant and subject: Violence against women (VAW) and attitudes. This is a very important and timely subject and is a potentially significant contribution to our field. Overall the article is well structured, written and represents a valid contribution to the literature in the field of VAW.

I just make two suggestions: i) the first is related to the fact that the introduction is too long, with too much information. I think it might be interesting to look at subtopics, such as Incidence of SSR, study objetives; ii) The second recommendation is related to the practical implications of the results obtained: how can the results help to design VAW prevention policies? The authors could develop this question further, in order to demonstrate greater practical applicability of their results

Author Response

Dear Reviewer,

According to your suggestions, we have revised and changed our paper “Street sexual harassment: experiences and attitudes among young Spanish people” (ijerph-1399638) in order to improve it.

Specifically we have made the following changes:

  • We have introduced subtopics in the Introduction section (1.1. Conceptualisation of Sexual Street Harassment (SSH), 1.2. The incidence of the Street Sexual Harassment (SSH), 1.3 The types of Street Sexual Harassment (SSH), 1.4. The perception of the Street Sexual Harassment (SSH), 1.5.The beliefs and attitudes towards Street Sexual Harassment (SSH), and 1.6. Study objectives).
  • We have developed our explanations about the practical implications of the results obtained.

We remain at your disposal for any further modifications or adjustments that you consider relevant.

Yours sincerely

The authors